# Quasi-Steady Analysis of a Small Wind Rotor with Swept Blades

**Mauro J. Guerreiro Veloso** [1,*] **, Carlos H. P. dos Santos** [2]**, Jerson R. P. Vaz** [1] **and Antonio M. Chaves Neto** [1]

1  Graduate Program in Natural Resources Engineering, Institute of Technology, Federal University of Pará, Belém 66075-110, PA, Brazil; jerson@ufpa.br (J.R.P.V.); amchaves@ufpa.br (A.M.C.N.)
2  Graduate Program in Mechanical Engineering, Institute of Technology, Federal University of Pará, Belém 66075-110, PA, Brazil; chpsantos@ufpa.br
*  Correspondence: mauroveloso@ufpa.br

**Abstract:** It is well known that wind power generation addresses the energy needs of small and remote populations as one of the alternatives to petroleum-based energy's greenhouse effect. Although there are several publications on rotor design and performance analysis, more should be written about the starting of wind turbines, mainly the small ones, where starting can be a big issue. The present paper evaluates the impact of the swept blade angle on the aerodynamic torque, thrust force, and minimal wind speed required to start the operation of a compact horizontal-axis wind turbine. It presents a novel investigation of the influence of swept rotor blades on the starting performance of a turbine drivetrain. The methodology uses the blade element moment theory coupled to Newton's second law, in which Palmgren's extended approach is employed. When the proposed methodology is compared to the experimental data available in the literature, it exhibits good agreement. However, when the wind turbine starts to run, the results show that swept blades do not always enhance the torque coefficient or reduce the thrust force as indicated in some scientific papers. For backward-swept blades, the maximum value decreases 4.0%. Similar behavior is found in thrust force for forward-swept blades. Therefore, more study is required to evaluate many blade foils in several operational environments to confirm this statement.

**Keywords:** wind turbine swept blade; small horizontal axis wind turbine; turbine starting

## 1. Introduction

In recent decades, the rise of industrial activity has boosted energy consumption [1]. It has mostly led to an increase in fossil energy sources, which has resulted in a rise in carbon emissions linked to global warming. According to statistics from the Brazilian Energy Balance 2022 [2], 78.07% of Brazil's energy supply comes from renewable sources, with wind energy accounting for 10.6%. As also reported by the Statistical Yearbook of Electricity [3], the increase of wind energy between 2020 and 2021 was 26.7%.

In addition to supplying electricity to extensive metropolitan areas, wind power facilities may also attend small settlements with energy [4,5]. This fact encourages the academic community's interest in studying small horizontal and vertical wind power turbines to provide electricity for low-energy-demand populations [6,7].

There are just a few studies on starting performance analysis of wind turbines. Rueda and Vaz [8] published an analysis of a turbine and generator's transient behavior in 2015. They apply the blade element theory, Newton's second law, and the Runge–Kutta technique of the fourth order to achieve this. Their results are in good agreement with experiments found in the literature. However, the methodology has a singularity in the vicinity of angular velocities equal to zero, which, according to the authors, makes it challenging to apply the method at turbine starting.

Kaya et al. [9] proposed an innovative swept-blade geometry design for a horizontal axis wind turbine. They analyzed turbine performance using computational fluid dynamics (CFD) techniques. Their outcomes show that swept-blade turbines have a power coefficient 2.9% higher than straight-blade ones, and for some cases, the thrust coefficient is 5.4% lower. They, also pointed out, that forward-swept-blade turbines enhance performance, but backward-swept-blade turbines reduce thrust force and, consequently, dissipation torque. Unfortunately, dissipative torque is disregarded in that study, and no turbine-starting evaluations are conducted.

Fritz et al. [10] proposed a correction model to extend the blade element momentum theory (BEMT) for swept blades. They reported that earlier studies had shown the effectiveness of swept blades using BEMT analysis. Its quick algorithm makes it suitable for evaluating numerous load cases in wind turbine certification. The correction model extends the methodology to account for the effects of swept blades, passively reduce loads, and optimize the design of wind turbine blades. They found good agreement between BEMT and the lifting line model regarding the local forces on the blades. However, the impact of the swept blades on the dynamic behavior of the turbine is not evaluated.

Vaz et al. [11] demonstrated a technique to assess the dissipative torque based on the Stribeck effects and Palmgren models to incorporate the static friction when the turbine starts from rest. Their model is validated with experimental measurements, leading to a good agreement between the experimental data and the theoretical model. In addition, the authors stated that the lowest evaluated wind speed required to start the turbine is 6.2% greater than the experimental wind tunnel measurements. Nevertheless, this study did not perform the effects of blade geometry changes on aerodynamic torque and turbine start performance.

A design methodology for high-capacity factor wind turbine applied to the Amazon is presented by Farias et al. [4]. Their study used the blade element theory and wind speed measurements in Salinópolis in the State of Pará, Amazon, to design the wind turbine. The numerical calculation revealed that the turbine's annual power capacity factor is equivalent to 22%, twice the performance of two commercial wind turbines. However, the nominal power designed turbine is less than the commercial ones. The outcomes show that the minimum estimated generating wind speed is 3.65 m/s, similar to the value determined by Vaz et al. [11]. The work revealed that the transient behavior had yet to be examined; hence, additional investigations are required for the turbine's start.

Celik [12] investigated the effect of the blades' number and turbine's moment of inertia on the performance of vertical-axis wind turbines (VAWTs) through CFD, which is validated by numerical and experimental data. The authors show that the change in moment of inertia did not impact the dynamic response of the turbine's starting and final rotation speeds. Nevertheless, as the number of blades grew, the minimum speed required to start the vertical turbine decreased. In addition, the investigation did not consider the bearings' dissipative forces, which are expected to impact the performance evaluation.

Moreira [13] performed an experimental investigation on the dissipative torque of a small horizontal-axis wind turbine (HAWT). The drivetrain resistance, using Palmgren and SKF models for bearing friction force are studied. The test bench outcomes agree closely with the theoretical proposed model. Furthermore, the authors assert that it can emulate small wind turbine performance in distinct regimes with different operation factors, power load generators, and dissipative loads on the drivetrain, which are design criteria for wind turbines; in addition, the author's statement highlights an investigation of turbine starting.

Hansen and Hansen [14] developed a comprehensive review on wind turbine noise generation, propagation, and their effects on humans and animals. They accurately estimate noise exposure applicable to large and medium scale wind farm and show a correlation between proximity to wind turbines and measures of discomfort and health-related quality of life. They comment on the importance of rotor with lower noise emission, which is a consequence of forward-swept blades. Another application of swept-blade modeling to large and medium scale turbines is investigated by Li et al. [15]. They proposed a

computational model applicable to turbines with swept blades under uniform inflow, perpendicular to the rotor. A good agreement with the BEMT method highlights the good performance of the method.

Pinheiro et al. [6] investigate the effect of dissipative torque generated by vertical-axis turbine ball bearings applying Newton's second law coupled with the double-multiple current tube method. Palmgren and SKF to determine the dissipative torque and the fourth-order Runge–Kutta to numerically evaluate the turbine's dynamic equation are also implemented. Nonetheless, the authors emphasize the necessity for more investigation on dynamic analysis during turbine starting to determine the turbine's behavior from quasi-steady to steady-state regimes.

Although there are several publications on the design and performance analysis of small horizontal-axis wind turbines [16,17], further investigation is required to examine the effects of swept blades on the starting and operational performance of small horizontal-axis wind turbines. The authors are unaware of any study on this regard. So, the present study evaluates how the swept-blade angle impacts the aerodynamic torque, thrust force, and the required wind speed for starting a small horizontal-axis wind turbine. In this case, Palmgren's extended method, blade element moment theory, and Newton's second law are employed in order to implement a quasi-steady model.

The investigation findings yield additional information regarding the dynamic behavior of the turbine during starting, including details on torque and angular velocity dependence on time. These factors are crucial for choosing the proper generator to attach to a wind turbine. Furthermore, this work also intends to add knowledge to the design and performance analysis of small wind turbines applied to small villages worldwide.

The remaining sections of this paper are arranged as follows. The next section exposes the turbine equation of motion, the blade element theory for swept-blade rotors, and the dissipative torque approach. Section 3 shows the outcomes and highlights the torque and thrust coefficients for distinct swept blades and suggestions for further investigations, and conclusion is explained in Section 4.

## 2. Theoretical Background

### 2.1. Newton's Second Law

The equation of motion for the turbine and generator set is obtained by assuming that the generator is coupled directly to the rotor, as illustrated in Figure 1. The wind turbine, generator, and shaft are rigid bodies, and the origin of the inertial coordinate system is fixed at the turbine center of mass $O$. Assuming $J_T$ and $J_g$ are the moment of inertia of the turbine and generator about the axis through the center of mass $O$, $\Omega$ is the angular velocity of the turbine and coupled generator. The resulting torques, $\sum T_i$, are about point $O$, acting on the turbine.

The equation of motion may be obtained by applying Newton's second law to the turbine and generator system and is written as follows:

$$\sum T_i = \left(J_g + J_r\right)\frac{d\Omega}{dt} \tag{1}$$

where

$\sum T_i$ is the sum of torques acting about axis through the turbine center of mass;

$\dfrac{d\Omega}{dt}$ is the turbine angular acceleration.

The left member of Equation (1) is written as

$$\sum_{i=1}^{n} T_i = T_r - T_D, \tag{2}$$

where

$T_r$ is the torque of aerodynamic force acting on the turbine around point O;

$T_D$ is the torque of dissipative forces acting on the turbine.

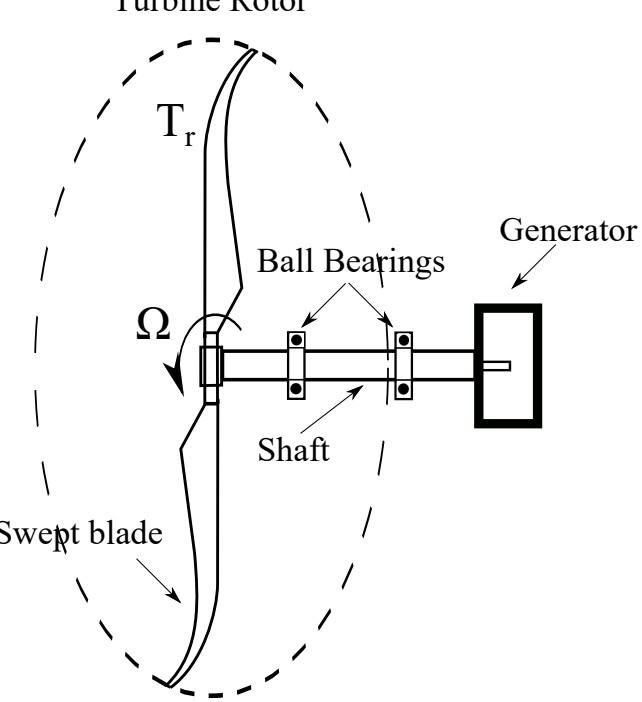

**Figure 1.** Illustration of the drivetrain for a wind turbine with swept blades.

Substituting expression (2) into Equation (1), it gives

$$\frac{d\Omega}{dt} = \frac{1}{(J_g + J_r)}(T_r - T_D).$$

(3)

The aerodynamic torque $T_r$ is performed by BEMT, which is described in Section 2.2. The dissipative torque is assumed to be the bearing friction forces and it is performed by a modified Palmgrem approach, which is thoroughly explained in refs. [11,13] and it is succinctly shown in Section 2.3. In this work, the Equation (3) can be solved by the fourth-order Runge–Kutta method starting from rest, knowing the elapsed time $\triangle t$ [18].

### 2.2. Blade Element Momentum for Rotors with Swept Blades

Here, BEMT is applied to evaluate the performance of a swept-blade turbine with distinct curved blades to derive performance characteristics at starting. The aerodynamic torque and thrust force are evaluated by blade element theory, where the turbine blade is divided into N elements with constant chord length $c_j$ and twist angle $\theta_j$. The aerodynamic torque is dependent on chord length, $c_j$; flow relative velocity, $W_j$; tangential and normal force coefficients, $C_{T,j}$ and $C_{N,j}$, respectively; the number of blades, $B$; and the radius $r_j$, as shown by BEMT approach.

The radius of turbine swept blades should be mapped dependent on the radius ratios and the local sweep angle $\beta_j$, in radians, and may be written as follow [19,20]

$$\Phi\left(\frac{r_j}{R}, \beta_j\right) = \left(\frac{r_j}{R}\right)^{1+\beta_j},$$

(4)

$$r_j = R\Phi\left(\frac{r_j}{R}, \beta_j\right), \forall\, j \in \{1, 2, ..., N\},$$

(5)

where $R$ is the tip radius of the swept blade and $\beta_j$ is the local angle of the swept blade which is written depending on the global angle of curvature, $\beta$.

Let $\beta_1 = 0°$ and $\beta_N = \beta$ in radians. Thus, in this paper, let $\beta_j$ be written as

$$\beta_j = \beta_1 + (j-1)\frac{(\beta_N - \beta_1)}{N-1} \, , \forall \, j \in \{1, 2, ..., N\}. \tag{6}$$

The swept blade turbine chord of the *jth* element is given by

$$c_j = c_j \cos(\beta_j). \tag{7}$$

At the small wind turbine starting, the exciter generator is turned off, preventing the system from producing power [21]. Hence, the turbine–generator set depends only on the kinetic energy of the wind to begin spinning, which balances the dissipative energy and inertia resistance [22]. The axial, $a$, and the tangential inductions factors, $a'$, are assumed null at starting [21], and the tip loss factors are neglected at BEMT parameter calculations [11,22].

In that order, at starting, the turbine can be modeled as quasi-steady state, and its assumption can be checked by the reduced frequency dimensionless parameter, $K_\alpha$, which is written as [11,21]:

$$K_\alpha \approx \frac{c}{2V_0(1+\lambda^2)^{3/2}}\left|\frac{d\lambda}{dt}\right|, \tag{8}$$

where $\lambda$ is the tip speed ratio written as follows:

$$\lambda = \frac{\Omega(t)R}{V_0(t)}. \tag{9}$$

The reduced frequency at tip values, $K_{\alpha,tip}$, that characterized flow [23] are

$$K_{\alpha,tip}\begin{cases} = 0, & \text{the flow is steady,} \\ \in \,]0, 0.05[, & \text{the flow is quasi-steady .} \\ \geq 0.05 & \text{the flow is unsteady} \end{cases} \tag{10}$$

To assess thrust and torque coefficients, the angle of attack $\alpha_j$ is necessary for each strip blade section, which is given by

$$\alpha_j = \phi_j - \theta_j \, , \forall \, j \in \{1, 2, ..., N\}, \tag{11}$$

where $\phi_j$ is the flux angle and $\theta_j$ is the blade twist angle, both in degrees.

The flow angle $\phi_j$ for swept-blade turbines is expressed as

$$\phi_j = \arctan\left[\frac{(1-a_{0,j})V_0}{\left(1+a_j'\right)\Omega r_j \cos(\beta_j)}\right] , \forall \, j \in \{1, 2, ..., N\}, \tag{12}$$

where $a_{0,j}$ and $a_j'$ are the axial and tangential induction factors, whose values should be approximately null at starting since the blades are stationary [11,21], and $\Omega$ is the turbine angular speed and $V_0$ is the wind speed, all for each '*jth*' blade section.

The normal and tangential force coefficients $C_{n,j}$, $C_{t,j}$ for swept blades are evaluated, respectively, for each $j$ section by [19]

$$C_{n,j} = (C_L \cos\phi_j + C_D \sin\phi_j)\cos\beta_j, \tag{13}$$

and

$$C_{t,j} = (C_L \sin\phi_j - C_D \cos\phi_j)\cos\beta_j, \tag{14}$$

where the values of lift, $C_L$, and drag, $C_D$, coefficients, shown in [11,19], depend on the angles of attack, $\alpha_j$, and Reynolds number.

The wind relative speed for values of $a \approx 0$ and $a' \approx 0$ is

$$W = \sqrt{V_0^2 + \left[\Omega r_j \cos(\beta_j)\right]^2}, \tag{15}$$

and the local solidity, $\sigma_j$, of each element section of a swept blade is

$$\sigma_j = \frac{B \cdot c_j}{2\pi r_j}. \tag{16}$$

The turbine thrust force and dissipative torque should be performed considering the rod between blade and hub, as illustrated in Figure 2. In this case, the aerodynamic characteristics of the rods are drag coefficient $c_d = 0.8$, radius at the rod tip $r_b = 0.143$, radius at the rod bottom $r_r = 0.055$, rod diameter $d_r = 0.00635$, and the following expressions to calculate thrust and torque:

$$A1 = \sqrt{V_0^2 + \Omega^2 r_b^2}, \tag{17}$$

$$A2 = \sqrt{V_0^2 + \Omega^2 r_r^2}, \tag{18}$$

$$A3 = 0.5\left(\Omega r_b A1 + V_0^2 \log\|\Omega r_b + A1\|\right), \tag{19}$$

$$A4 = 0.5\left(\Omega r_r A2 + V_0^2 \log\|\Omega r_r + A2\|\right), \tag{20}$$

$$A5 = V_0^2 + 2r_r^2 \Omega^2, \tag{21}$$

$$C1 = r_b V_0^2 A1 + 2r_b^3 \Omega^2 A1 - r_r A2 A5, \tag{22}$$

$$C2 = V_0^4 \left[\sinh^{-1}(r_r \Omega / V_0) - \sinh^{-1}(r_b \Omega / V_0)\right]. \tag{23}$$

In this work, the small wind turbine measured by [11] is used. The blades and rotor were designed to reproduce the experimental rotor of Weegeref [24], who used circular arc airfoils of constant chord. The present blades had a chord of 4 cm and a twist of 46° at their inner edge at radius 14.3 cm and 17° at the tip radius of 34.0 cm. The lift and drag coefficients of these curved airfoils are experimentally determined by Bruining [25] over a range of Reynolds numbers, Re, and angles of attack up to 90°. Further description of the wind rotor can be found in [11,19].

The thrust force $F_T$ and aerodynamics torque $T_r$ evaluated by the BEMT model are given, respectively, by

$$F_T(t) = \frac{1}{2}\rho B \left[\int_{r_h}^{R} W^2(r) \cdot c(r) \cdot C_n(r)\, dr + d_r c_d \frac{V_0}{\Omega}(A3 - A4)\right], \tag{24}$$

and

$$T_r(t) = \frac{1}{2}\rho B \left[\int_{r_h}^{R} W^2(r) \cdot c(r) \cdot C_t(r) \cdot r\, dr - \frac{d_r c_d}{8\Omega^2}(\Omega C1 + C2)\right]. \tag{25}$$

Equations (24) and (25) are used to calculate the dissipative torque, $T_D$, through Palmgren's approach. They are substituted into Equation (3) and solved numerically by fourth-order Runge–Kutta to obtain $\Omega$. From the tip speed ratio, $\lambda$, an expression for $\Omega$ may be as

$$\Omega = \frac{V_0 \lambda}{R}. \tag{26}$$

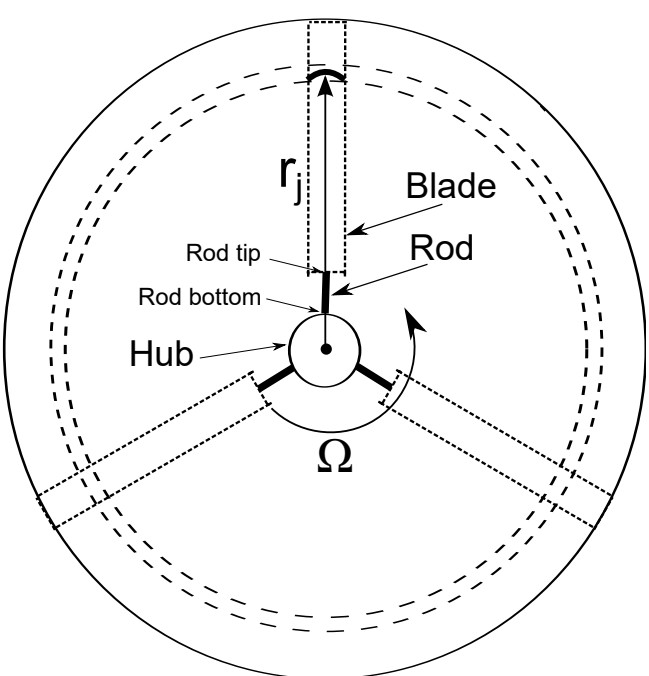

**Figure 2.** Illustration of the small wind turbine simulated.

Using (26) in expressions (27) and (28), it yields

$$A1 = V_0 \sqrt{1 + \left(\frac{\lambda r_b}{R}\right)^2},$$ (27)

and

$$A2 = V_0 \sqrt{1 + \left(\frac{\lambda r_r}{R}\right)^2}.$$ (28)

Replacing $\Omega$, $A1$ and $A2$ by identity (26), (27), and (28) in $A3$, $A4$, and $A5$, respectively, leads to

$$A3 = 0.5 V_0^2 \left[ \frac{\lambda r_b}{R} \sqrt{1 + \left(\frac{\lambda r_b}{R}\right)^2} + \log \left\| V_0 \left[ \frac{\lambda r_b}{R} + \sqrt{1 + \left(\frac{\lambda r_b}{R}\right)^2} \right] \right\| \right],$$ (29)

$$A4 = 0.5 V_0^2 \left[ \frac{\lambda r_r}{R} \sqrt{1 + \left(\frac{\lambda r_r}{R}\right)^2} + \log \left\| V_0 \left[ \frac{\lambda r_r}{R} + \sqrt{1 + \left(\frac{\lambda r_r}{R}\right)^2} \right] \right\| \right],$$ (30)

and

$$A5 = V_0^2 \left[ 1 + 2 \left(\frac{\lambda r_r}{R}\right)^2 \right].$$ (31)

So then, the expressions for B1, B2, B3, B4, D1, and D2 are

$$B1 = \sqrt{1 + \left(\frac{\lambda r_b}{R}\right)^2},$$ (32)

$$B2 = \sqrt{1 + \left(\frac{\lambda r_r}{R}\right)^2},$$ (33)

$$B3 = \left[ 1 + 2 \left(\frac{\lambda r_b}{R}\right)^2 \right],$$ (34)

$$B4 = \left[ 1 + 2\left(\frac{\lambda r_r}{R}\right)^2 \right], \tag{35}$$

$$D1 = 0.5\rho B d_r c_d \frac{V_0}{\Omega}(A_3 - A_4), \tag{36}$$

and

$$D2 = \frac{\rho B d_r c_d}{16\Omega^2}(\Omega C1 - C2). \tag{37}$$

Now, substituting (26), $A3$ and $A4$ in $D1$, it gives

$$D1 = \frac{0.25\rho B d_r c_d R V_0^2}{\lambda}\left[ \frac{\lambda}{R}(r_b B1 - r_r B2) + \log\left|\frac{\lambda r_b + RB1}{\lambda r_r + RB2}\right| \right]. \tag{38}$$

So, the thrust force can be expressed by

$$F_T = \frac{\pi\rho R^2}{\lambda^2}\int_{x_h}^{\lambda} W^2 \sigma C_n x\, dx + D1, \tag{39}$$

where $x = \Omega r / V_0$ is the local speed ratio and $x_h = r_{hub}\Omega / V_0$. Thus, the thrust coefficient is given by

$$C_T = \frac{2}{\lambda^2}\int_{r_h}^{\lambda}\left(\frac{W}{V_0}\right)^2 \sigma C_n x\, dx + \frac{0.5 B d_r c_d}{\pi R \lambda}\left[ \frac{\lambda}{R}(r_b B1 - r_r B2) + \log\left|\frac{\lambda r_b + RB1}{\lambda r_r + RB2}\right| \right]. \tag{40}$$

Substituting expression (26), A5, B1, B2, B3, and B4 into C1, C2 gives

$$D2 = \frac{\rho B d_r c_d R^2 V_0^2}{16\lambda^2}\left[ B1 B3 \frac{\lambda r_b}{R} - B2 B4 \frac{\lambda r_r}{r} + \sinh^{-1}\left(\frac{\lambda r_r}{R}\right) - \sinh^{-1}\left(\frac{\lambda r_r}{R}\right) \right]. \tag{41}$$

Hence, the torque coefficient may be written as

$$C_Q = \frac{2}{\lambda^3}\int_{x_h}^{\lambda}\left(\frac{W}{V_0}\right)^2 \sigma C_t x^2\, dx + D3, \tag{42}$$

where

$$D3 = \frac{B d_r c_d}{8\pi R \lambda^2}\left[ B1 B3 \frac{\lambda r_b}{R} - B2 B4 \frac{\lambda r_r}{r} + \sinh^{-1}\left(\frac{\lambda r_r}{R}\right) - \sinh^{-1}\left(\frac{\lambda r_r}{R}\right) \right]. \tag{43}$$

### 2.3. The Dissipative Torque

In this work, the dissipative torque is performed through the Palmgren approach [11]. An empirical formulation combined with the Stribeck model to account for the static frictional force when the turbine begins to rotate is employed [22]. Figure 3 depicts the dissipative torque, showing its variation over time at the turbine starting from rest. In Figure 3, in the beginning, static friction torque shows a constant linear function (blue line) at a really short time; after that, the friction torque turns down abruptly, which is the Stribeck effect, and the turbine starts (red line).

The torque increases slowly, representing the turbine acceleration, and so the dynamic frictional torque is much smaller than the static friction one (black line). This phenomenon is important because it shows that the aerodynamic torque needs to be big enough to overcome the resistive torque from the bearings.

The extended Plamgren approach [11,13] is written as

$$T_{D,P*}(t) = T_L + T_V + T_S \exp\left[ -\left(\frac{n}{n_{st}}\right)^i \right] + 0.5 C_{MPB}, \tag{44}$$

where $T_L$ is the load-dependent frictional torque performed by Equation (45), $T_V$ is the viscous friction torque evaluated by Equation (49), $T_S$ is the static friction torque, Equation (50), $C_{MPB}$ is the drag torque constant of the encoder and the magnetic particle brake model MPB70, and $i$ and $n_{st}$ are parameters determined experimentally in [11] by regression analysis as listed in Table 1.

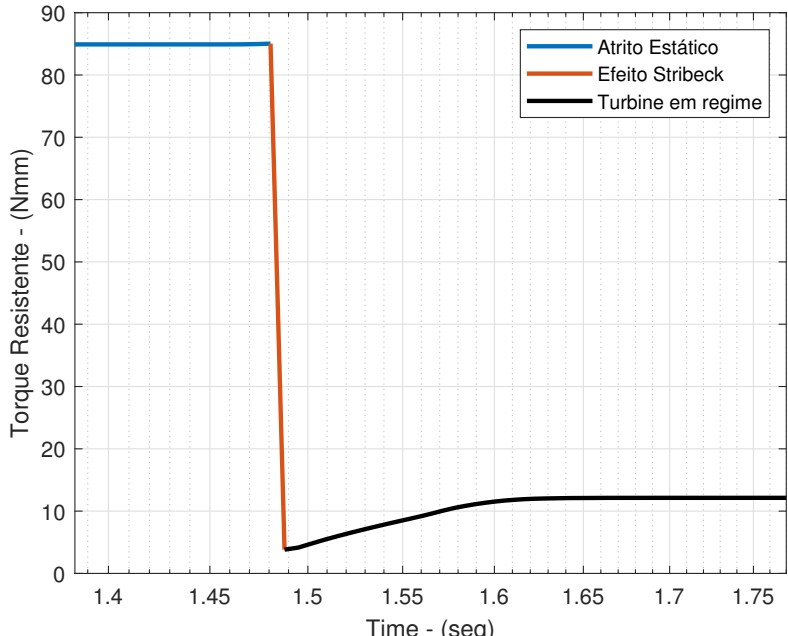

**Figure 3.** Dissipative torque pattern of small horizontal axis wind turbine at starting.

**Table 1.** Parameter values applied to modified Palmgren's model for radial deep-groove ball bearings [11].

| Parameter | Description | Values |
|:---:|:---:|:---:|
| $C_s$ | Basic static-load rating | 4.0 kN |
| $X_s$ | Radial load factor | 0.6 |
| $Y_s$ | Axial load factor | 0.5 |
| $z$ | Load empiric factor to Equation (46) | 0.0004 |
| $y$ | Exponent empiric factor to Equation (46) | 0.55 |
| $d_m$ | Bearing pitch diameter | 31 mm |
| $i$ | Stribeck exponent | 0.26 |
| $n_{St}$ | Stribeck parameter | |
| $C_{MPB}$ | Drag torque constant | 113 Nmm |

Thus, the parameter $T_L$ is written as

$$T_L = f_L F_\beta d_m, \tag{45}$$

where $d_m$ is the pitch diameter available in Table 1, and $f_L$ is a parameter depending on the rolling-element bearing type and is it evaluated by the following expression [11,13]

$$f_L = z \left( \frac{F_s}{C_s} \right)^y, \tag{46}$$

where $C_S$, z, and y are parameters found in Table 1. The static equivalent load $F_S$ is given by

$$F_S = X_s F_r + Y_s F_T, \tag{47}$$

where $X_s$, $Y_s$ are parameters found in manufacturer's catalogue, these values are given in Table 1; $F_r$ is the resultant of radial loads applied on the bearing, and $F_T$ is the thrust load carried out from BEMT.

The $F_\beta$ is performed by

$$
F_\beta = \begin{cases} \max[0.9F_T \cot(\alpha) - 0.1F_r, F_r], & \text{for radial bearing} \\ 3F_T - 0.1F_r, & \text{for deep-groove ball bearing, and } \alpha = 0. \\ F_T, & \text{for thrust bearing} \end{cases} \quad (48)
$$

The viscous friction torque for moderate speed, $T_V$ in Nm, is given by Palmgren empirical equation in the form

$$
T_V = \begin{cases} 10^{-7} f_0 (\nu_0 n)^{2/3} d_m^3, & \text{if } \nu_0 n \geq 2000 \\ 160 \times 10^{-7} f_0 d_m^3, & \text{if } \nu_0 n < 2000 \end{cases} \quad (49)
$$

where $\nu_0$ is the cinematic viscosity, whose value is 315.6 cSt; $n$ is the numerical solution of (3) in revolutions per minute and $f_0$ for deep-groove ball bearings and is equal to 2 when lubricated in an oil bath. References [6,11] give different values for $f_0$ associated with bearing types and lubrication methods.

The static friction torque for rolling bearing is assumed to be equal to mean average between the sliding friction torque and friction torque seal [11], and it is written as

$$
T_S = \frac{1}{2}(T_{sl} + T_{seal}), \quad (50)
$$

where the sliding frictional torque $T_{sl}$ is computed by equation

$$
T_{sl} = G_{sl}\mu_{sl}. \quad (51)
$$

For deep-groove ball bearings, $G_{sl}$ is performed by

$$
G_{sl} = S_1 d_m^{-0.145} \left( F_r^5 + \frac{S_2 d_m^{1.5} F_T^4}{\sin(\alpha_F)} \right), \quad (52)
$$

where $S_1$ and $S_2$ are parameters values given in Table 2, while the parameter $\alpha_F$ is given by the following expression

$$
\alpha_F = 24.6 \left( \frac{F_T}{C_S} \right)^{0.24}. \quad (53)
$$

The sliding friction coefficient $\mu_{sl}$, in Equation (51) of sliding frictional torque, is evaluated by

$$
\mu_{sl} = \phi_{bl}\mu_{bl} + (1 - \phi_{bl})\mu_{EHL}, \quad (54)
$$

where the parameters $\mu_{bl}$ and $\mu_{EHL}$ are listed in Table 2. The mixed lubrication weighting factor $\phi_{bl}$ is carried out by

$$
\phi_{bl} = \left[ -2.6 \times 10^{-8} (\nu_0 n)^{1.4} d_m \right]. \quad (55)
$$

The friction torque caused by the rolling bearing seal type in the bearing is

$$
T_{seal} = K_{S1} d_S^{\beta_*} + K_{S2}, \quad (56)
$$

where the factors $K_{S1}$, $K_{S2}$, and $\beta_*$ are dependent on the bearing type, and the $d_S$ is the shoulder rolling bearing diameter, whose value is given in Table 2.

Additional and thorough comprehensive information about the expanded Palmgren method is found in [11,13].

**Table 2.** Parameter values applied to the sliding $T_{sl}$ and seal $T_{seal}$ frictional torque [11].

| Parameter | Description | Values |
|:---:|:---:|:---:|
| $S_1$ | Sliding factor | $4.63 \times 10^{-3}$ |
| $S_2$ | Sliding facto | 4.25 |
| $\mu_{bl}$ | Cofficeint dep. on additivies | 0.15 |
| $\mu_{EHL}$ | Friction coeff. full film bearing | 0.15 |
| $K_{S1}$ | Bearing type constant | 0.018 |
| $K_{S2}$ | Bearing type constant | 0.0 |
| $d_s$ | Roller bear. shoulder diam. | 42 mm |
| $\beta_*$ | Exponent dep. on bearing seal | 2.25 |

## 3. Results and Discussion

In this section, the numerical solution of the quasi-static model is compared to the experimental data available in [11]. The model assumed the number of blade sections equals 30, and lift and drag coefficient values are correlated to 60,000 Reynolds numbers [25]. Calculations based on the BEMT model assumed quasi-steady behavior, in which axial and tangential induction factors $a$ and $a'$ are equal to zero, as explained in Section 2.2. The blade was subdivided into 30 blade elements to calculate aerodynamic torque. Calculations are performed without tip loss to be congruent with the assumption of the null induction factor.

The aerodynamic (25) and dissipative torque (44) expressions are introduced into the equation of motion (3). The resulting expression is solved numerically by the fourth-order Runge–Kutta method, which uses a time step of 0.5 s, an overtime equal to 60 s, and an initial condition for the angular velocity of $1.0 \times 10^{-10}$ rad/s.

The semicircular airfoil three blades turbine of 0.34 m tip radius and 0.040 m of constant chord is illustrated and described in [11]. At the starting turbine measurement, the magnetic brake MPB70 model, coupled to the shaft turbine, is also employed. The blade twist angle, $\theta$, changes from 46° at the inner edge to 17° at the turbine blade tip [11]. Figure 4 illustrates the variation of the twist angle $\theta$ and the local component chord $c_i$ over the dimensionless ratio $r/R$. The detailed input data and conditions for the numerical model are shown in Table 3.

**Table 3.** Detailed data for the numerical model.

| Parameter | Description | Value |
|:---:|:---:|:---:|
| $r_b$ | Hub radius | 0.143 m |
| $R$ | Tip blade radius | 0.34 m |
| $N_b$ | Number of radius elements | 30 |
| $B$ | Number of blades | 3 |
| $c$ | Chord length (constant) | 0.04 m |
| $\theta$ | Twist angle | $[17°, 46°]$ |
| $J_T$ | Total mass moment of inertia | 0.0991 kgm$^2$ |
| $\beta$ | Swept-blade angle in degree | $-30, -20, -10, 0, 10, 20, 30$ |
| $C_L$ | Lift coefficient | from [25] |
| $C_D$ | Drag coefficient | from [25] |
| $rho$ | Air density | 1.205 kg/m$^3$ |
| $\nu$ | Kinematic viscosity | $1.511 \times 10^{-5}$ N-s/m$^2$ |
| $T$ | Time range | $[0, 60]$ s |
| $\Delta t$ | Time step | 0.5 s |
| $\Omega_0$ | Initial angular velocity | $1.0 \times 10^{-10}$ rad/s |
| $V_0$ | Wind speed | $[0, 6, 30]$ m/s |
| $T_d$ | MPB70 drag torque not excited | 113 Nmm |
| $N$ | Max number of iteration | 200 |
| $\epsilon$ | error | 0.0001 |

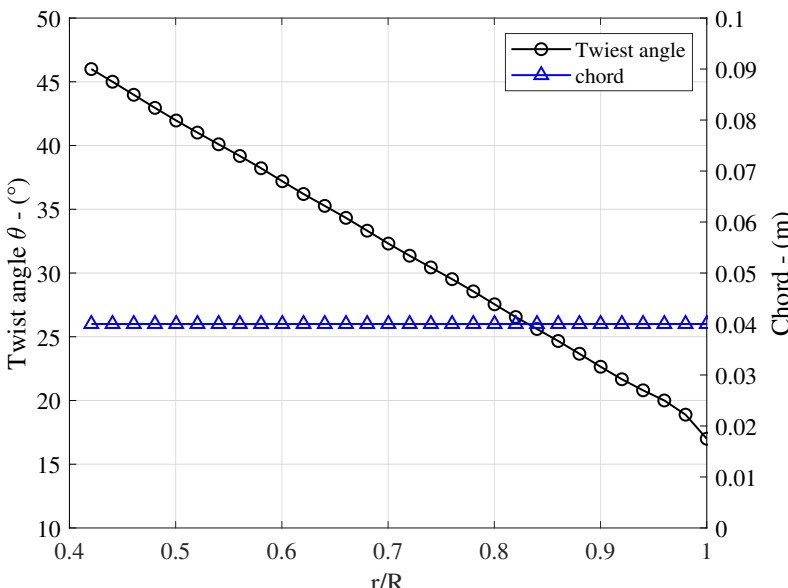

**Figure 4.** Twist angle and chord distribution over dimensionless ratio r/R.

Figure 5 shows the straight blade geometry for $\beta$ equal to $0°$, which is the rotor geometry experimented by [11]. Such a geometry is compared to forward and backward blade rotors. Figure 6a depicts the forward-swept blade for $\beta$ equal to $-30°$. The dimensions of the forward rotor are sketched in Figure 6b, in which the tip radius, $R$, hub radius, $r_h$, and root radius, $r_r$, as well as its direction rotation are shown. Figure 7a shows the backward turbine swept blade, $\beta$ equal to $30°$. Additionally, the dimensions of the backward rotor are shown in Figure 7b.

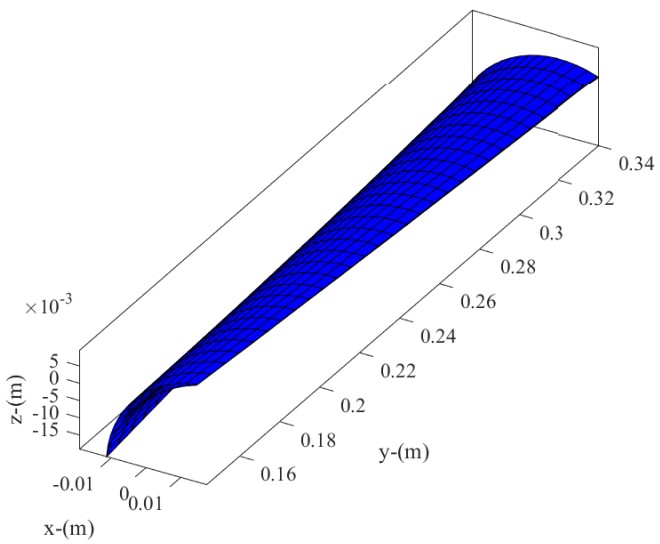

**Figure 5.** Straight blade shape, clockwise turbine rotation.

The quasi-steady state is verified by the reduced frequency parameter, $K_\alpha$, and Equation (8) performed for the straight-blade ($\beta = 0°$), for the swept blade ($\beta = 30°$), and for the straight blade measurements reported by Vaz et al. [11]. The calculated values are compared in Figure 8, which indicates that the highest reduced frequency levels for turbine straight blades measured and theoretical data are $6.21 \times 10^{-4}$ and $5.48 \times 10^{-4}$, respectively. At the same time, the maximum reduced frequency value for swept blades with $\beta = 30°$ is $5.33 \times 10^{-4}$. All these numerical quantities are in $]0.0, 0.05[$, Equation (10), which confirms the assumption of quasi-steady regime, and, as reported by [11,21,23],

the reduced frequency values are very small to alter the lift and drag coefficients. These reduced frequency parameters are depicted in Figure 8.

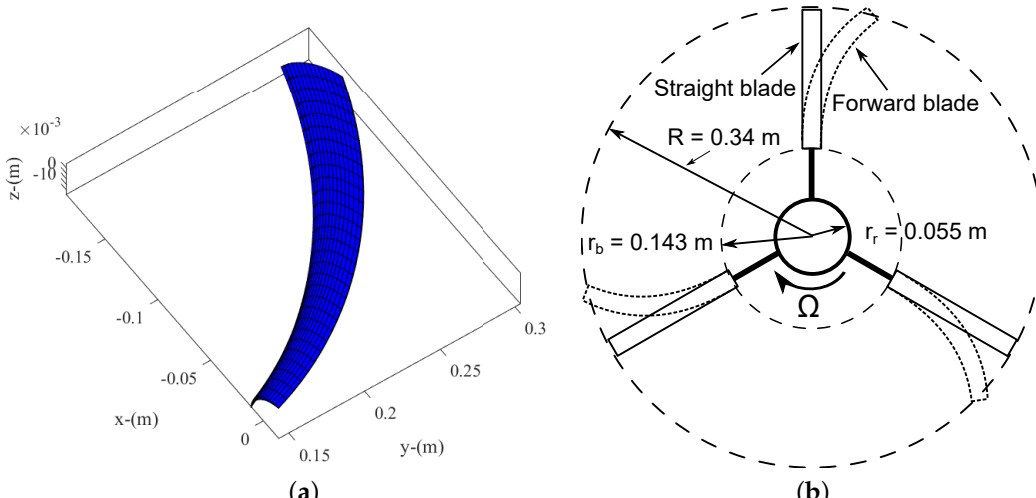

**Figure 6.** (**a**) Swept blade shape for −30°, clockwise turbine rotation. (**b**) Forward blade sketch and rotation direction.

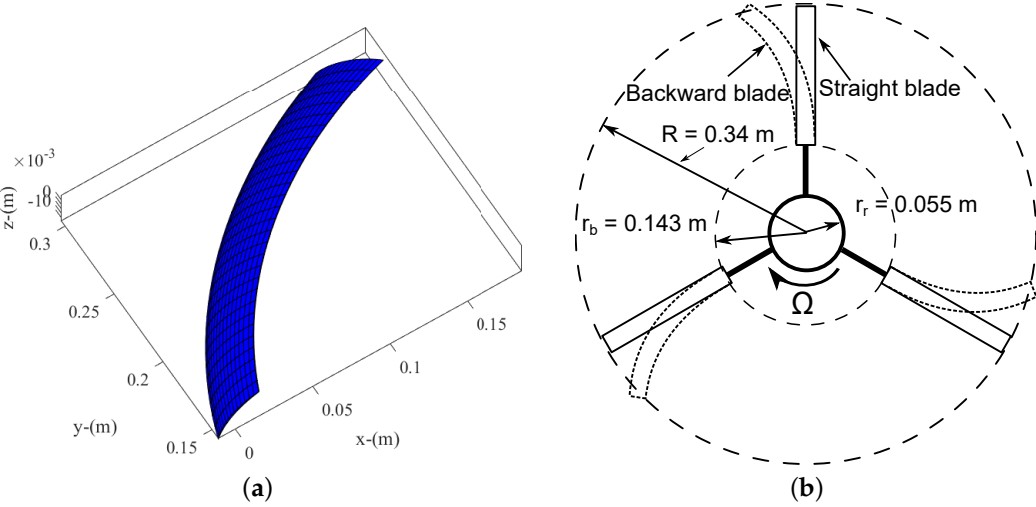

**Figure 7.** (**a**) Swept blade shape for 30°, clockwise turbine rotation. (**b**) Backward blade sketch and rotation direction.

The numerical simulation results are compared to the measurements made at the University of Calgary in the Schulich School of Engineering's Aero-Energy Wind Tunnel Laboratory [11], which is 7.6 m long, with a contraction ratio of 5.76, and an open working section of 1 square meter, reaching a maximum wind speed of 19 m/s.

Table 4 shows the time discretization number and the relative error between the mean numerical angular velocity and the mean experimental measurements evaluated over a steady-state time range ( 32 s ≤ time ≤ 60 s). The table shows the accuracy between the numerical simulation and experimental data regarding angular velocity.

**Table 4.** Number of discretization time, angular velocity average $\overline{\Omega}$ (rad/s), and error (%).

| Number of Time Steps | $\overline{\Omega}_{experimental}$ | $\overline{\Omega}_{numerical}$ | Error |
|---|---|---|---|
| 40 | 35.915 | 35.391 | 1.48 |
| 80 | 35.902 | 35.392 | 1.44 |
| 120 | 35.859 | 35.372 | 1.37 |

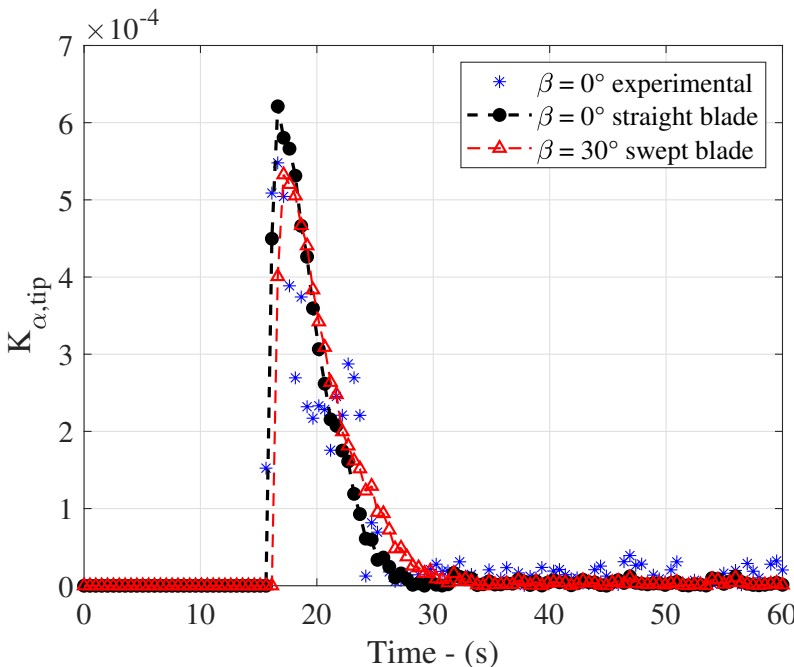

**Figure 8.** Reduced frequency parameter of the turbine blade tip over time.

Figure 9 depicts the numerical solution of expression (3). It shows the results for the angular speed, $\Omega$, compared with the measured values as well as the estimated net torque Equation (2), and the wind speed measured for the straight blade. At runaway, the numeric angular speed average is 35.368 rad/s, and the average of the angular speed measured is 35.845 rad/s, with an error of 1.33%. The net torque curve for straight blades is also displayed. Note that the net torque reaches the maximum value in the unsteady condition. This is because, at starting, the wind velocity variation increases the net torque a little soon after turbine starting, decreasing it rapidly as the wind velocity reaches a constant value.

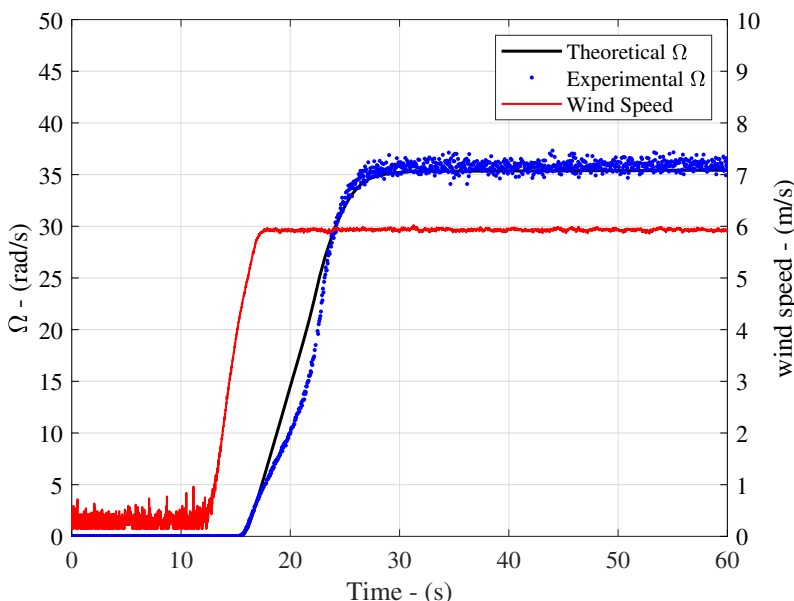

**Figure 9.** Experimental and theoretical angular speed over time for various blade geometries, adapted from Vaz [11].

Figure 10 illustrates how the torque coefficient $(C_Q)$ changes with differing swept-blade angles $(\beta_i)$ and tip speed ratios $(\lambda)$. It reveals that these values increase as the sweep angle shifts from $-30°$ for forward curved blades to $0°$ for straight blades; after

that, the maximum values begin to decrease as the sweep angle shifts from 0° to 30° for backward curved blades However, for $\lambda$ greater than 1.4, the corresponding coefficient values are more significant than the straight blades ones. These findings contradict the conclusion reported by Kaya [9] that turbines using forward blades are more efficient than straight blades.

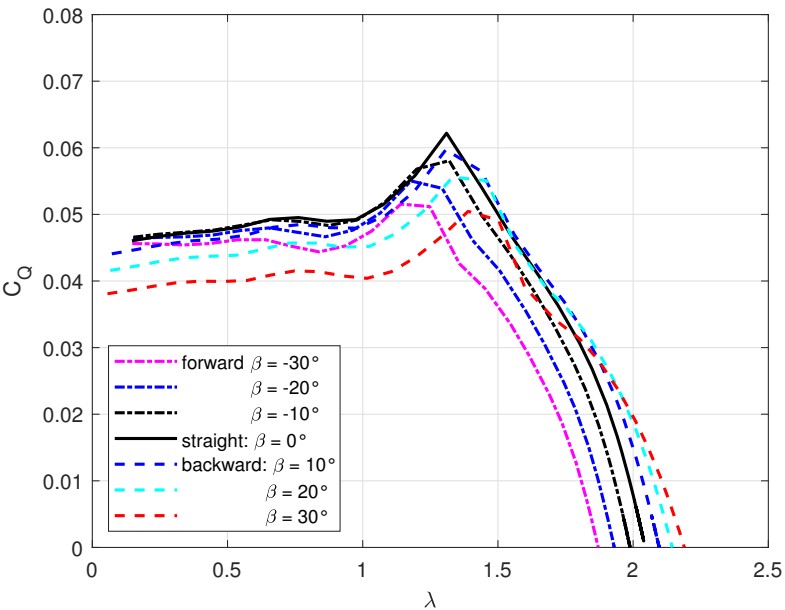

**Figure 10.** Torque coefficient over tip speed ratio for distinct blade sweep angles.

However, the geometric swept blade, defined by Kaya [9], is based on the radial point of the blade's curvature starting and the blade's transverse tip distance between the curved and straight blade. The ratio $C_L/C_D$ of the blades applied by Kaya [9] is 11 times the ratio utilized in this research, resulting in a superior aerodynamic performance. Furthermore, its turbine model is on-power operation. The results reported by Gemaque [20] are consistent with this work's conclusion, in which at starting there is no velocity induction at the rotor blades, leading to a no-power extraction condition. Nevertheless, the optimal performance for a sweeping blade angle is 30 degrees for backward-curved blades, and the turbine is in operation.

Figure 11 shows the change in torque coefficient with time for different curved blade angles. The graph indicates that straight blades have the highest peak torque, but the peak torque of swept blades falls as the forward or backward angle increases. It is also shown that the turbines with the smallest peak torque coefficients are for the backward angle of 30°. This analysis follows the preceding finding about the torque coefficient over the $\lambda$ graph shown in Figure 10. These results indicate that the foil energy conversion of arc circle swept blades is less efficient than that of straight blade turbines, except for the 10 degrees of the backward blade. To generalize this remark is necessary to investigate the impact of some other distinct foil shapes with various swept-angle blades on starting performance. Additionally, arc circle blades seems to have complex behavior of the boundary layer detachment on the airfoil at low Reynolds number.

Figure 12 shows how thrust coefficients change based on sweep angle and tip speed ratio. The lowest thrust coefficient corresponds to the −30 degrees, and all forward blades show a thrust coefficient less than straight blades. However, for the backward blade, the thrust is more significant than for a straight blade after reaching its maximum value. For $\beta = 10°$, the $C_T$ values are approximately equal to that of straight blade before the maximum point. These findings indicate that some swept blade may reduce the thrust coefficient under particular operational conditions [20], which contradicts the Kaya [9] conclusions as previously described.

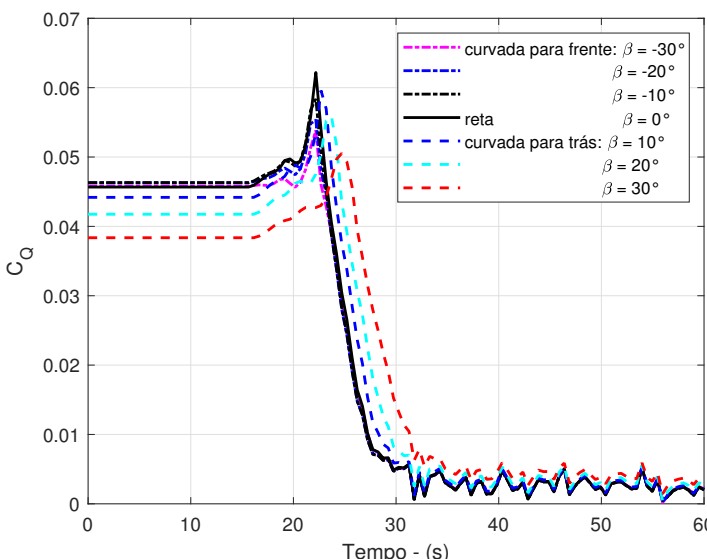

**Figure 11.** Torque coefficients for different blade sweep angles, $\beta$, with time.

Figure 13 displays the thrust coefficient over time for blades with different sweep angles. After reaching the maximum value of the thrust coefficient, the backward-swept blades have higher values than the straight blades, which is about 24.78% higher than the straight blades. In addition, for $\beta$ equal to $-30°$, forward-swept blade, the thrust coefficient is up to 27.2%, lower than straight blades.

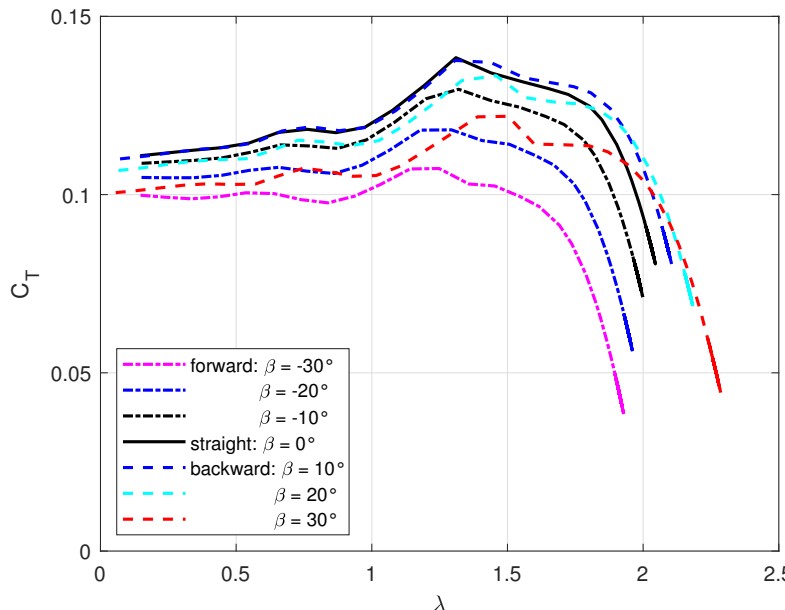

**Figure 12.** Thrust coefficient over tip speed ratios for different blade sweep angles.

Figure 14 exhibits the net torque over time achieved numerically through the aerodynamic torque Equation (25) and the extended Palmgren's expression (44) into Equation (2). The graph shows that forward-curved blades have a shorter period than backward ones, resulting in a lower time variance needed to reach steady-state. In addition, the graph reveals that backward-curved blades have a broader time range, leading to faster-rated speeds, as shown in Figure 15. Around 32 s, there is no torque, and the angular velocity stays roughly constant.

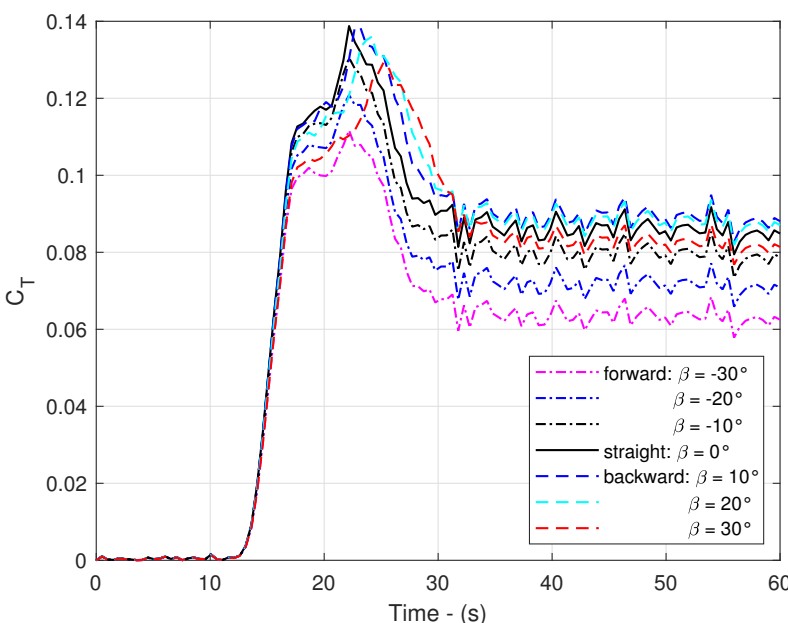

**Figure 13.** Thrust coefficient over time for different blade sweep angles.

Figure 15 depicts wind and angular velocity for various swept-angle turbine blades over time. At 32 s, the net torque of both straight and forward-curved blades approaches zero, as seen in Figure 14, and the angular speed is almost constant. In addition, the forward-swept blades have the slowest runaway speed, suggesting that they generate less energy.

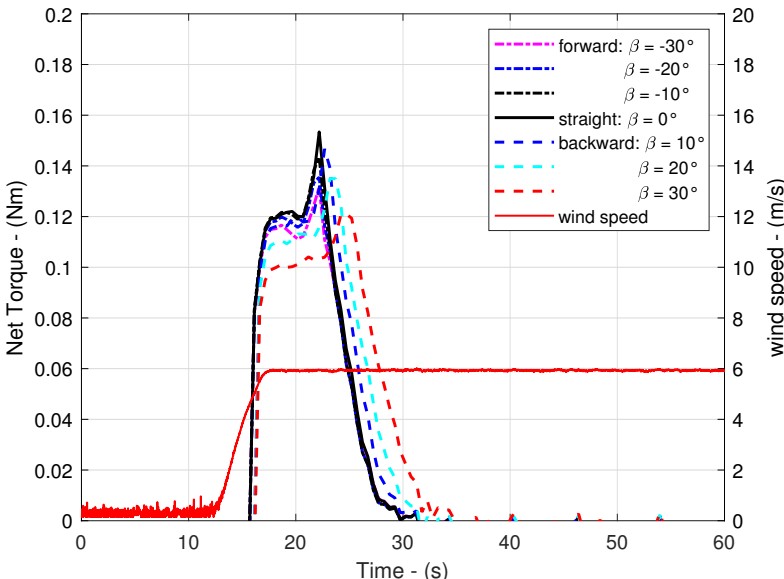

**Figure 14.** Net torque over time variation for various sweep angle configurations for turbine blades.

Figure 16 shows that the minimum wind speed to start forward-swept blades, for $\beta$ angles equal to $\{-30°, -20°, -10°, 0°\}$, is approximately 4.55 m/s, and 5.039 m/s for the backward-swept blade to $\beta$ equal to $\{10°, 20°, 30°\}$. These results show an increase of 10.7% in wind speed starting with a change from forward- to backward-turbine-swept blades, corresponding to a net torque greater for straight and forward-swept blades than backward-swept blades (Figure 14). For a certain blade curvature angle, the thrust required can decrease, and the aerodynamic torque can increase. It has significant implications for understanding the design and performance of turbines. Here, the turbine with forward-swept blades starts faster than that with backward-swept blades.

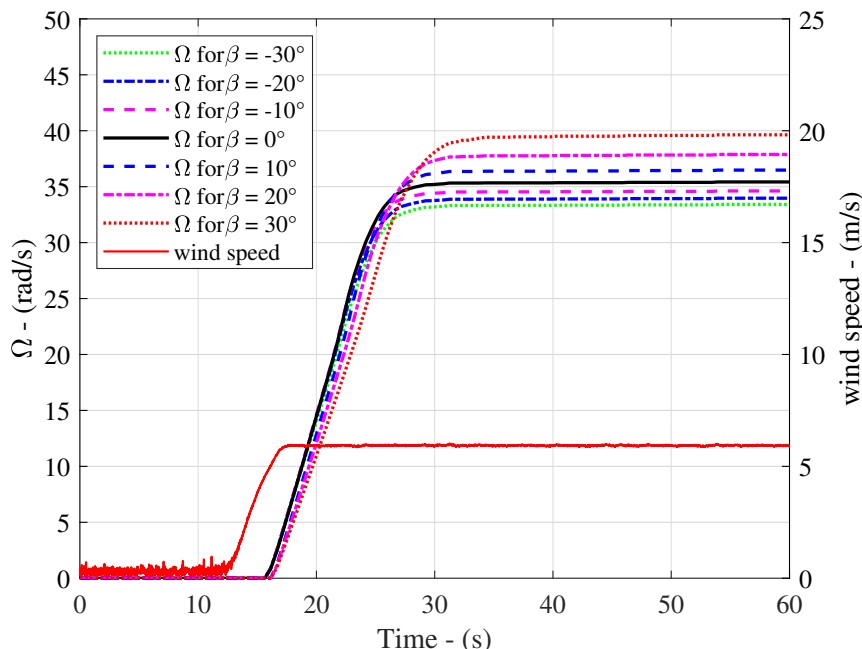

**Figure 15.** Angular and experimental wind speed over time for distinct swept-blade angles, *β*.

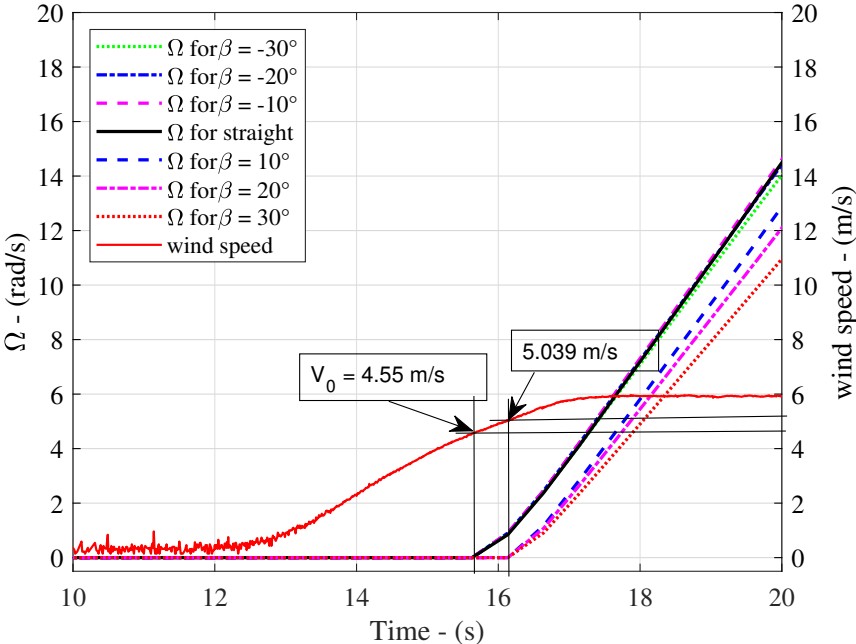

**Figure 16.** Wind speed for swept turbine starting with different swept blade angle, adapted from [11].

Table 5 shows the aerodynamic torque, mean angular velocity, $\overline{\Omega}$, and average angular acceleration, $\Delta\Omega/\Delta t$, values obtained from the time simulation between 0 and 60 s with a 0.5 s time step associated with each blade curvature configuration. These data show that for turbines with forward-curved blades and semicircular profiles, the average acceleration values are greater than for turbines with backward-curved blades, suggesting that the elapsed time between the start of motion and the steady state time is shorter than for backward blades, indicating that these turbines are faster-starting units. Hence, these turbines exhibit quicker reactions to load changes while being charged.

Table 5 also shows that the maximum torque of the straight blade rotor is higher than these with curved blades. Only the turbine with backward blades ($\beta = 30°$) showed the lowest peak value, as in Figure 14. This can be attributed to the lower thrust coefficients, as in Figures 12 and 13. These data suggest that turbines with straight and forward-curved

blades present shorter starting times and lower final speeds when under steady state (see the angular speed in Figure 15, and the data in Table 5). As a consequence, lower angular speeds lead to lower dynamic forces at the rotor blades and lower noise emissions. These results can be expanded for medium and large turbines with effects on noise generation, as shown by Hansen and Hansen [14]. Usually, the optimization models available in the literature concentrate in rated parameters to design optimum blades; however, it is necessary to analyze the starting condition, and swept blades play an important role in this regard.

**Table 5.** Maximum torque evaluations, mean acceleration data, and average angular velocity for different swept-blade turbine designs with a semicircular profile.

| Parameter | $\beta = -30$ | $\beta = -20$ | $\beta = -10$ | $\beta = 0$ | $\beta = 10$ | $\beta = 20$ | $\beta = 30$ [1] |
|---|---|---|---|---|---|---|---|
| Torque (Nm) | 0.132 | 0.135 | 0.143 | 0.153 | 0.146 | 0.135 | 0.121 |
| $\Delta\Omega/\Delta t$ (rad·s$^{-2}$) | 3.39 | 3.51 | 3.59 | 3.63 | 3.58 | 3.39 | 3.01 |
| $\overline{\Omega}$ (rad·s$^{-1}$) | 33.29 | 33.89 | 34.54 | 35.54 | 36.37 | 37.70 | 39.24 |

[1] $\beta < 0$ forward blade, $\beta = 0$ straight blade, and $\beta > 0$ backward blade.

## 4. Conclusions

This study investigated the impact of swept blades on the starting performance of a small wind turbine. According to the results, swept blades may not lower thrust or enhance torque, depending on the operating condition. These findings contradict the conclusion reported by Kaya [9] that turbines using forward blades are more efficient than straight blades. Actually, it depends on the aerodynamic characteristics of the airfoil used. Additionally, it suggests an optimal sweep angle that can minimize thrust force and boost torque coefficient and is an important contribution to understanding the design of rotors with swept blades. Moreover, it supports the selection of an appropriate electrical generator to be coupled to a wind rotor. The quasi-steady algorithm ensures applicability for turbine size as well as it being fast and easy to implement in any computer, and it is extendable even to turbines with a diffuser.

Another important conclusion is on the use of circular arc airfoil with a constant chord in the turbine rotor. The thrust with swept blades is not always less than the thrust of turbines with straight blades when using curved plate airfoils. This seems to be due to the complex behavior of the boundary layer detachment on the airfoil at low Reynolds numbers. Thus, additional investigation on different airfoil shapes and curves is necessary to confirm or not these results.

**Author Contributions:** Conceptualization, M.J.G.V. and J.R.P.V.; data curation and methodology, M.J.G.V., C.H.P.d.S. and J.R.P.V.; software, M.J.G.V. and J.R.P.V.; validation, M.J.G.V. and J.R.P.V.; formal analysis, M.J.G.V., J.R.P.V., C.H.P.d.S. and A.M.C.N.; writing—original draft preparation, M.J.G.V., C.H.P.d.S., J.R.P.V. and A.M.C.N.; writing—review and editing, M.J.G.V., J.R.P.V. and A.M.C.N.; supervision, J.R.P.V. and A.M.C.N. All authors have read and agreed to the published version of the manuscript.

**Funding:** This research received no external funding.

**Informed Consent Statement:** Not applicable.

**Data Availability Statement:** Not applicable.

**Acknowledgments:** The authors would like to thank the CNPq, PDPG-CAPES. (Agreement: 88881.707312/2022-01), and PROPESP/UFPA (PAPQ) for the support.

**Conflicts of Interest:** The authors declare no conflict of interest.

## Nomenclature

The following symbols are used in this manuscript:

| | |
|---|---|
| $a, a'$ | Axial and tangential induction factor |
| $A1, A2, A3, A4, A5$ | Temporary parameter |
| $B$ | Number of blades |
| $B1, B2, B3, B4, B5$ | Temporary parameters |
| c | Chord |
| $C_D$ | Drag coefficient |
| $C_L$ | Lift coefficient |
| $C_n$ | Normal force coefficient |
| $C_t$ | Tangential force coefficient |
| C1, C2 | Temporary parameters |
| $C_T$ | Thrust coefficient |
| $C_Q$ | Torque coefficient |
| $C_S$ | Basic static-load rating |
| $C_{MPB}$ | Drag torque cosntant |
| $d_m$ | Bearing pitch diameter |
| $d_s$ | Roller bearing shoulder diameter |
| $F_r$ | Radial load |
| $F_T$ | Thrust load |
| $F_S$ | Static equivalent load |
| $F_\beta$ | Palmgrem empirical parameter |
| $G_{sl}$ | Empiric slidding friction load |
| i | Stribeck exponent |
| $J_T$ | Turbine mass moment of inertia about center of mass |
| $J_g$ | Generator mass moment of inertia |
| $K_{\alpha,tip}$ | Tip reduced frequency |
| $K_1, K_2$ | Slide bearing friction parameter |
| n | Frequency rotation in rpm |
| $n_{st}$ | Stribeck frequency rotation in rpm |
| $r_j$ | Radial position of blade section |
| $r_h$ | Hub radius |
| R | Turbine radius |
| S1, S2 | Sliding parameters |
| $T_{D,P*}$ | Extended Palmgren approach bearing dissipative torque |
| $T_L$ | Load friction torque |
| $T_V$ | Viscous friction torque |
| $T_r$ | Aerodynamic torque |
| $T_{sl}$ | Sliding rolling bearing friction torque |
| $T_{seal}$ | Friction torque on seals rolling bearing |
| $V_0$ | Freestream wind speed |
| $X_s$ | Rolling bearing radial load factor |
| $Y_s$ | Rolling bearing axial load factor |
| $y$ | Load empiric factor |
| $W$ | Flow relative speed |
| $z$ | Exponent load factor |
| Greek Symbols | |
| $\alpha$ | Angle of attack |
| $\alpha_F$ | Rolling bearing angular contact |
| $\beta$ | Blade sweep angle |
| $\beta_*$ | Exponent factor for empiric seal friction |
| $\theta$ | Blade twist angle |
| $\lambda$ | Tip speed ratio |
| $\mu_{bl}$ | Slide friction bearing parameter |
| $\mu_{EHL}$ | Friction coefficient for full film |
| $\phi_{bl}$ | Mixed lubrication weighting factor condition |
| $\phi_{ish}$ | Inlet shear heating reduction factor |

| $\phi_{rs}$ | Kinematic replenishment/starvation reduction factor |
| $\Omega$ | Turbine angular speed |

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
