# Peer review of "Quasi-Steady Analysis of a Small Wind Rotor with Swept Blades"

_sustainability, doi:10.3390/su151310211_

Round 1

Reviewer 1 Report

The work presented is an interesting study as well as containing intriguing conclusions. The authors of the study closely correspond with previous studies and also build on and use the results of studies by other authors.

Despite the relatively detailed description of the investigated rotor blades, it would be worthwhile to include in the paper a sketch showing the described angles in the rotor blade, so that the reader would not have to guess how the blade is arranged and which angle the authors are writing about.

(2) It would also be worthwhile to present in more detail the analysis itself, there is no information on the computational domain and the boundary conditions adopted, the parameters of the analysis.

3) I also found no information on the conditions under which the measurement/analysis was carried out.

4. there is also no information about the model itself - what grid, what level of convergence was assumed.

5. the paper cites the results of an earlier paper by one of the authors, were the presented results obtained for an identical model?

Author Response

Detailed Response to Reviewer 

We are pleased to resubmit the revised version of sustainability-2434324 “Quasi-steady Analysis of a Small Wind Rotor with Swept Blades”. We appreciated the constructive comments and criticisms of the Reviewers. We have addressed each of their concerns as outlined below. In addition, some corrections in English language have been made to improve the clarity of the manuscript.  

Reviewer #1 - Comments:

1) The work presented is an interesting study as well as containing intriguing conclusions. The authors of the study closely correspond with previous studies and also build on and use the results of studies by other authors. Despite the relatively detailed description of the investigated rotor blades, it would be worthwhile to include in the paper a sketch showing the described angles in the rotor blade, so that the reader would not have to guess how the blade is arranged and which angle the authors are writing about.

Answer: The authors thank the reviewer for the suggestion. Figures 6b and 7b were included in the manuscript to clarify the dimensions used for forward and backward blades. Also, table 3 with all parameters used in the simulation was added.

2) It would also be worthwhile to present in more detail the analysis itself, there is no information on the computational domain and the boundary conditions adopted, the parameters of the analysis.

Answer: Thank you for your comment. The Quasi-static model, Runge-Kutta of fourth order to solve the Newton’s second law and table 3 with all parameters and input data are included in the text.

3) I also found no information on the conditions under which the measurement/analysis was carried out.

Answer: Thank you for your comment. The information about measurements and parameter conditions was included in table 3.

4) there is also no information about the model itself - what grid, what level of convergence was assumed.

Answer: Sensibility analysis for different time step was added in table 4.

  1. the paper cites the results of an earlier paper by one of the authors, were the presented results obtained for an identical model?

Answer: The authors appreciate the comment. The earlier paper Gemaque et al. [19] developed an optimization model only for backward rotor blades. They do not deal with forward blades; they concentrate on the new optimization of chord and twist angle distributions. The present work evaluates the impact of the swept blade angle on the aerodynamic torque, thrust force, and minimal wind speed required to start the operation of a compact horizontal-axis wind turbine. It presents a novel investigation into the influence of swept rotor blades on the starting performance of a turbine drivetrain. This was added in the abstract.

Author Response

Detailed Response to Reviewer

 We are pleased to resubmit the revised version of sustainability-2434324 “Quasi-steady Analysis of a Small Wind Rotor with Swept Blades”. We appreciated the constructive comments and criticisms of the Reviewers. We have addressed each of their concerns as outlined below. In addition, some corrections in English language have been made to improve the clarity of the manuscript.

Reviewer #2 - Comments:

First of all, I would like to appreciate the authors for their effort; there are some modifications which would help if incorporated for better understanding to readers.

Answer: Thank you for the constructive comments to improve our paper.

1) Authors are clear about the problem specification and same is not reflected in the abstract.

Answer: Thank you very much for your comment. The problem specification was included in the abstract.

2) Authors need to add more standard literature survey on the wind turbine blade designing and their specifications, which would help the readers to understand the concept better.

Answer: Thank you for your suggestion. Papers on swept blades are very difficult to find in the literature. Also, we are unaware of any study on the effect of swept blades to turbine drivetrain. However, as suggested by the reviewer, standard literature survey was included in the introduction, and two more references was added.

  1. Hansen, C.; Hansen, K. Recent Advances in Wind Turbine Noise Research. Acoustics 2020, 2, 171–206. https://doi.org/10.3390/ 473acoustics2010013.

  1. Li, A.; Pirrung, G.R.; Gaunaa, M.; Madsen, H.A.; Horcas, S.G. A computationally efficient engineering aerodynamic model for swept wind turbine blades. Wind Energy Science 2022, 7, 129–160. https://doi.org/10.5194/wes-7-129-2022.

3) What is the scope of this swept blades if this is used for the medium/large scale turbine. Mention if there is any specific reason and through some light in the literature section this will further improve the readers interest, may through this point they can have some clarification

Answer: Thanks for this suggestion. In the Introduction (lines 82-91) a text on this regard was included.

4) Proper draft for the tables and spellings are not prepared properly check for the table 2, Fig. 4 title twist, in figure twist

Answer: Thank you, it was corrected in the text.

5) Results and discussion section is not convincing need to be developed more by comparing with straight, forward, and backwards data then it would be clear for understanding and also, data taken is from real-time or through simulation? need to be confirmed in paper.

Answer: Thank you for your suggestion. Results and discussion are improved in the text, between figure 16 and the conclusion, which shows the comparison of outcomes from backward, straight, and forward blades, and comment that this geometry configuration is applied to medium and large wind turbine and reduce emission noise from those turbines. All the results are evaluated from simulated time.

6) For further improvisation of the paper other factors like de-rating and relative parameters need to be considered when designing a turbine blade to check how effectively this can be employed in real-time applications

Answer: An improved discussion was included in the manuscript. As stated in question 5, the data depicted in table 5 improve the discussion, demonstrating that the maximum torque of the straight blade rotor is higher than the curved ones, which is attributed to the lower thrust coefficients. These data suggest that turbines with straight and forward-curved blades present shorter starting times and lower final speeds when under steady state. Usually, optimization models available in the literature concentrate in rated parameters to design optimum blades, however it is necessary to analyze the starting condition, and swept blades plays an important role on this regard.
